# Dynamics of Urine Metabolomics and Tubular Inflammatory Cytokines in Type 1 Diabetes Across Disease Durations

**DOI:** 10.3390/metabo15110734

**Published:** 2025-11-10

**Authors:** Mei-Shiuan Yu, Chih-Yung Chiu, Fu-Sung Lo, Wei-Cheng Lin, Li-Jia Wu, Cih-Yi Yen, Mei-Ching Yu

**Affiliations:** 1Department of Microbiology and Immunology, School of Medicine, Tzu-Chi University, Hualien 970374, Taiwan; meishiuan@gms.tcu.edu.tw; 2College of Medicine, Chang Gung University, Taoyuan 333323, Taiwan; pedchest@cgmh.org.tw (C.-Y.C.); lofusu@cgmh.org.tw (F.-S.L.); 3Department of Pediatric Pulmonology, Lin-Kou Chang Gung Memorial Hospital, Taoyuan 333423, Taiwan; 4Clinical Metabolomics Core Laboratory, Lin-Kou Chang Gung Memorial Hospital, Taoyuan 333423, Taiwan; 5Department of Pediatric Endocrinology and Genetics, Lin-Kou Chang Gung Memorial Hospital, Taoyuan 333423, Taiwan; 6Department of Electrical Engineering, Chang Gung University, Taoyuan 333323, Taiwan; weiclin@mail.cgu.edu.tw; 7Department of Pediatric Nephrology, Lin-Kou Chang Gung Memorial Hospital, Taoyuan 333423, Taiwan; ricawoo@cgmh.org.tw (L.-J.W.); ufo11140113@cgmh.org.tw (C.-Y.Y.); 8Department of Chemical Engineering, Ming Chi University of Technology, New Taipei City 243303, Taiwan

**Keywords:** Type 1 diabetes, diabetic kidney disease, amino acid metabolomics, urinary tubular biomarkers

## Abstract

**Background/Objectives**: Type 1 diabetes (T1D) is a chronic autoimmune disease characterized by sustained inflammation, leading to diabetic kidney disease (DKD). This study investigated urinary tubular injury biomarkers and metabolomic profiles in relation to albuminuria and renal function across varying durations of T1D. **Methods**: A cross-sectional analysis was conducted in 247 youth-onset T1D patients categorized by disease duration: short ≤ 5 years (T1D-S, n = 62), medium 6–10 years (T1D-M, n = 67), and long > 10 years (T1D-L, n = 118). Urinary cytokines (MCP-1, KIM-1, NGAL) were measured by ELISA. Metabolomic profiling was performed using ^1^H-NMR spectroscopy. **Results**: Urinary MCP-1/Cr, KIM-1/Cr, and NGAL/Cr levels were significantly elevated in T1D patients compared with non-diabetic controls, but did not correlate with disease duration. Metabolomic profiling identified distinct urinary signatures across T1D duration. Specifically, N-acetylcysteine (NAC) and N-delta-acetylornithine (NAO) increased progressively, while N-acetylaspartate (NAA) and pyruvic acid decreased with longer disease duration. These four metabolites remained statistically significant after both based on Mann–Whitney tests with false discovery rate (FDR) correction (q < 0.05) and application of a conservative alpha threshold (*p* < 0.01), suggesting potential disruptions in amino acid and carbohydrate metabolism. **Conclusions**: Urinary biomarkers (MCP-1/Cr, NGAL/Cr, and KIM-1/Cr) are sensitive indicators of subclinical kidney dysfunction in T1D patients, often preceding albuminuria. Alterations in amino acid-related metabolites (NAC, NAA, and NAO) and pyruvate highlight possible metabolic disturbances associated with T1D duration and oxidative stress. However, given the cross-sectional design, longitudinal studies are needed to confirm causality and clarify their predictive value in DKD progression.

## 1. Introduction

Type 1 diabetes (T1D) is a chronic inflammatory disorder characterized by the destruction of insulin-producing pancreatic beta cells, leading to hyperglycemia and subsequent metabolic disturbances that affect multiple organs and tissues [1,2]. The global incidence of T1D continues to rise, particularly among children and adolescents, representing significant public health challenges [3,4]. Importantly, the associated physical and psychological comorbidities negatively impact long-term patient outcomes.

Diabetic kidney disease (DKD) is a major complication of T1D, frequently leading to chronic kidney disease (CKD) and end-stage kidney failure (ESKF) [5,6]. Its pathogenesis involves complex interactions between hyperglycemia, hypertension, and genetic susceptibility, causing glomerulosclerosis, interstitial inflammation and fibrosis, and arteriolar hyalinosis [7,8,9].

Compared to adults, youth with diabetes exhibit a more aggressive clinical progression than adults, with accelerated loss of β-cell function, insulin resistance, and early onset of organ damage [10,11,12]. These may be attributed to differences in hormonal milieu, growth-related metabolic demands, and potential challenges in achieving optimal glycemic control during childhood and adolescence. Thus, early detection of renal dysfunction is critical for preventing progression to ESKF and achieving better patient outcomes. Currently, DKD diagnosis relies primarily on serum creatinine-based estimated glomerular filtration rate (eGFR) and albuminuria [13]. However, these markers may not reliably detect early kidney damage or predict long-term disease outcomes, especially since reduced renal function can occur even without albuminuria [14,15,16]. Therefore, more sensitive and specific biomarkers for early kidney injury in T1D are urgently needed.

Recent advances in urinary biomarkers and metabolomics offer promising diagnostic alternatives for DKD [17,18]. Biomarkers such as kidney injury molecule-1 (KIM-1), neutrophil gelatinase-associated lipocalin (NGAL), and monocyte chemoattractant protein-1 (MCP-1) have demonstrated potential in identifying early tubular injury and inflammation [19,20,21]. Furthermore, urine metabolomic analyses further provide insights into metabolic disturbances preceding clinical manifestations of DKD.

This study aims to investigate the multifaceted relationships between urine metabolomics, inflammatory mediators, albuminuria, and renal functions across different disease durations in T1D patients. By characterizing the metabolic and inflammatory profiles in T1D patients at various disease stages, we aim to identify potential biomarkers of early kidney dysfunction and diabetic progression, which may help understand the pathophysiological mechanisms underlying diabetic kidney injury, particularly in youth-onset T1D patients.

## 2. Materials and Methods

### 2.1. Study Protocol and Subjects

We conducted a prospective, cross-sectional study at the Pediatric Tertiary Care Center of Lin-Kou Chang Gung Memorial Hospital, Taiwan, from March 2017 to January 2018. The study enrolled 247 patients diagnosed with childhood-onset T1D according to World Health Organization criteria. Patients were grouped by disease duration: short (T1D-S, ≤5 years, n = 62), medium (T1D-M, 6–10 years, n = 67), and long (T1D-L, >10 years, n = 118). Furthermore, 60 subjects served as non-diabetic controls for comparison of urinary cytokine levels with T1D patients. These individuals were initially referred to the pediatric nephrology clinic for evaluation of suspected urinary abnormalities (e.g., abnormal urine test results or voiding dysfunction such as urinary frequency). Following a comprehensive assessment—including repeat urinalysis, blood tests for kidney function, and renal/bladder ultrasonography—all control subjects were confirmed to have normal renal function and no structural abnormalities.

Clinical parameters, including age, age at diagnosis, gender, body mass index (BMI), systolic blood pressure (SBP), and diastolic blood pressure (DBP), along with biochemical markers, including fasting plasma glucose (FPG), glycohemoglobin (HbA1c), total cholesterol (TC), low-density lipoprotein cholesterol (LDL-C), high-density lipoprotein cholesterol (HDL-C), triglyceride (TG), uric acid, high-sensitivity C-reactive protein (hs-CRP), homocysteine, blood urea nitrogen (BUN), and eGFR were recorded. Hematological parameters (white blood cells (WBC), hemoglobulin (Hb), and platelets and urine albumin to creatinine ratio (UACR) were also documented. Morning voided urine samples were collected and centrifuged at 2000 rpm for 10 min at 4 °C. Subsequently, the urinary supernatants were stored at −80 °C until enzyme-linked immunosorbent assay (ELISA) and metabolomics analyses.

### 2.2. Quantitative ELISA Analysis of Urinary MCP-1, KIM-1, and NGAL

Urinary MCP-1, KIM-1, and NGAL were measured using the Human CCL2/MCP-1 DuoSet ELISA Kit (catalog no. DY279-05), Human TIM-1/KIM-1/HAVCR DuoSet ELISA kit (catalog no. DY1750B), and Human Lipocalin-2/NGAL DuoSet ELISA kit (catalog no. DY1757) from R&D Systems, Minneapolis, MN, USA. These measurements were made in accordance with the manufacturer’s instructions. The assay sensitivities were 15.6 pg/mL for MCP-1 and KIM-1, and 78.1 pg/mL for NGAL. Chemokine levels were expressed as ng/mg creatinine.

### 2.3. ^1^H-Nuclear Magnetic Resonance (NMR) Spectroscopy Analysis

Urine samples were prepared and analyzed according to a previously described protocol [22]. Briefly, urine samples (900 μL) were mixed with phosphate buffer in deuterium water (100 μL) containing 0.04% TSP (3-(trimethylsilyl)-propionic-2,2,3,3-d4 acid sodium salt) for chemical shift standardization. After centrifugation at 12,000× *g* for 30 min at 4 °C, 600 μL of the supernatant was analyzed using a Bruker Avance 600 MHz NMR spectrometer (Bruker-Biospin GmbH, Karlsruhe, Germany). Spectral preprocessing was performed with TopSpin v3.2 (Bruker BioSpin, Rheinstetten, Germany) and NMRProcFlow v1.4 [23], including baseline correction, chemical shift calibration, solvent peak exclusion, and spectral alignment to minimize variability due to pH, ionic strength, or temperature differences. Spectral quality was assessed by signal-to-noise ratio, and noisy or inconsistent regions were excluded. To account for concentration differences and reduce potential batch effects, bucket intensities were normalized before multivariate analysis.

For spectral data reduction, both intelligent bucketing and variable-size (adaptive) binning were applied [24]. Intelligent bucketing allows bin edges to adapt to local spectral features, reducing the risk of splitting narrow peaks or merging adjacent peaks. Variable-size binning further refines this process by recursively adjusting bin boundaries according to peak shape, thereby minimizing artifacts from small chemical-shift variations and improving alignment across spectra. The combined use of these methods ensured optimal capture of biologically relevant spectral information.

Metabolite identification was performed using Chenomx NMR Suite v8.6. Data were logarithmically transformed, mean-centered, and Pareto-scaled prior to analysis in MetaboAnalyst v5.0. Multivariate analyses included partial least squares discriminant analysis (PLS-DA), with metabolites having variable importance in projection (VIP) scores ≥ 1.0 considered significant contributors to group separation.

### 2.4. Metabolite Differential Analysis and Pathway Evaluation

Differences in metabolites between groups were assessed using Mann–Whitney tests (MetaboAnalyst). To account for multiple comparisons, *p*-values were adjusted using the false discovery rate (FDR) method, and adjusted q-values were reported. For pairwise comparisons of individual metabolites, features with raw *p* < 0.05 were initially identified, and the robustness of these findings was further validated through Random Forest modeling combined with Boruta feature selection and 20-fold cross-validation. In addition, correlations between urinary metabolites, biochemical parameters, and urinary cytokines were examined using Spearman’s correlation analysis (R software v4.1). Venn diagrams were generated using the matplotlib-venn and matplotlib packages in Python 3, while pathway diagrams were created using networkx (v3.1) and matplotlib (v3.7.1).

### 2.5. Statistical Analyses

Descriptive statistics were expressed as medians with interquartile ranges (IQR). Normality of continuous variables was assessed using the Shapiro–Wilk test, and many variables demonstrated significant deviations from normality (Appendix A). Therefore, comparisons across subgroups and by albuminuria status were performed using Kruskal–Wallis ANOVA followed by Dunn’s post hoc tests. Binary logistic regression was applied to assess associations between urinary cytokines and albuminuria, adjusting for covariates including HbA1c, age, sex, BMI, eGFR, and diabetes duration. Pearson correlation coefficients were also calculated. For metabolomic data, *p*-values from Mann–Whitney tests were adjusted for multiple testing using the Benjamini–Hochberg FDR method, with adjusted q < 0.05 considered statistically significant. In addition, to further reduce the risk of false discovery from multiple testing, statistical significance was also conservatively defined as *p* < 0.01. All tests were two-tailed. Analyses were conducted using IBM SPSS Statistics version 20.0 (IBM Corp., Armonk, NY, USA) and OriginPro 2023 (OriginLab Corp., Northampton, MA, USA).

## 3. Results

### 3.1. Demographic and Clinical Profiles of T1D Groups

The study included 247 youth-onset T1D participants (121 males, 126 females; median age 18.6 years [IQR 13.6–22.3]), divided by disease duration: short (T1D-S, n = 62; median age 12.5 years [IQR 9.3, 16.2]), medium (T1D-M, n = 67; median age 16.9 years [IQR 13.2, 20.6]), and long (T1D-L, n = 118; median age 22.0 years [IQR 19.0, 25.9]). Clinical and laboratory characteristics of the entire cohort and the three distinct groups are summarized in Table 1 and Table 2. As anticipated, the age of diagnosis, disease duration, blood pressure (SBP, DBP), and BMI increased with T1D duration, and were higher in the T1D-L group, reflecting their older age. Hb levels were within normal age-specific ranges but lower in the T1D-S group, likely reflecting age differences. Levels of HbA1c, hs-CRP, TC, HDL-C, TG, eGFR, UACR, WBC, and platelet counts did not significantly differ between the three groups. However, LDL-C was significantly elevated in the T1D-L group. Although uric acid and homocysteine remained within normal ranges, both increased progressively with T1D duration (Pearson correlation in Appendix A).

### 3.2. Urinary MCP-1/Cr, KIM-1/Cr, and NGAL/Cr Levels in T1D Patients

Urinary cytokines (uMCP-1/Cr, uKIM-1/Cr, uNGAL/Cr) were significantly elevated in T1D patients compared with non-diabetic controls, regardless of disease duration (Table 2). This finding supports the notion that diabetes is a chronic inflammatory condition with potential long-term adverse effects on kidney health. Cytokine levels did not correlate with BMI, which remained within normal age-specific ranges across all groups. Interestingly, patients with long-duration T1D appeared to have lower uMCP-1/Cr concentrations than those with shorter disease duration (*p* < 0.05).

Furthermore, urinary cytokine levels were also significantly higher in T1D patients with albuminuria (UACR ≥ 30 mg/g) compared with those without albuminuria (Table 3). However, in binary logistic regression analyses adjusted for HbA1c, age, sex, BMI, eGFR, and diabetes duration, urinary cytokines (uMCP-1/Cr, uKIM-1/Cr, and uNGAL/Cr) were not independently associated with albuminuria (all *p* > 0.05). Although urinary cytokine levels were higher in albuminuric patients, their independent predictive value diminished in multivariable models after adjustment for HbA1c and other covariates, reflecting confounding and collinearity rather than a protective effect. In contrast, HbA1c consistently demonstrated a strong association with albuminuria, with odds ratios ranging from 0.63 to 0.67 (*p* < 0.001). The overall models were statistically significant (likelihood ratio χ^2^ *p* < 0.0001) and correctly classified approximately 90% of cases (Appendix A).

These findings indicate that glycemic control, as reflected by HbA1c, remains the principal determinant of albuminuria in this cohort, whereas urinary cytokines did not independently predict albuminuria status after adjustment for covariates. Additionally, no significant sex differences in urinary cytokine levels were observed (Appendix A).

### 3.3. Distinct Urinary Metabolic Phenotypes and Functional Pathways of T1D-S, T1D-M, and T1D-L Patients

Based on the VIP ≥ 1.0 and fold-change criteria, metabolomic analysis revealed significant differences among T1D duration groups (Table 4). Fourteen characteristic urinary metabolites—including N-acetylcysteine (NAC), N-acetylaspartic acid (NAA), N-acetylornithine (NAO), acetylsalicylate, phenylacetylglycine (PAGly), dimethylamine, creatine, 2-aminobutyric acid, pyruvic acid, N6-acetyllysine, formic acid, tyrosine, pantothenic acid, and leucine—were significantly altered between T1D-L and T1D-S patients. When comparing T1D-S to T1D-M, eight metabolites (NAC, NAA, NAO, acetylsalicylate, PAGly, dimethylamine, creatine, and 2-aminobutyric acid) showed significant differences, while five (NAC, NAA, NAO, pyruvic acid, and N6-acetyllysine) differed between T1D-M and T1D-L groups.

As illustrated in the Venn diagram (Figure 1), three metabolites—NAC, NAA, and NAO—were consistently altered across all comparisons, underscoring their potential relevance to T1D progression. Importantly, all key metabolites (NAC, NAA, NAO, and pyruvic acid) satisfied both the FDR-adjusted threshold (q < 0.05) and the conservative alpha level (*p* < 0.01), supporting the robustness of these findings.

Subsequent pathway enrichment analysis indicated disruptions primarily in amino acid and carbohydrate metabolism (Table 5). Pyruvic acid emerged as a central metabolite involved in interconnected pathways such as alanine, aspartate, and glutamate metabolism, as well as glyoxylate and dicarboxylate metabolism (Figure 2), highlighting its integrative role in metabolic reprogramming during T1D progression.

### 3.4. Relationships Between Urinary Cytokines (uKIM-1/Cr, uMCP-1/Cr, uNGAL/Cr) and Metabolites

Pearson correlation analysis revealed significant associations among urinary metabolites, tubular injury biomarkers, renal function indices, and systemic biochemical parameters, as illustrated in the heatmap (Figure 3). It demonstrated that uKIM-1/Cr ratios correlated positively with urine metabolites such as PAGly, creatine, pyruvic acid, and NAC. The uMCP-1/Cr ratio correlated with pyruvic acid, reflecting carbohydrate metabolism disturbances linked to renal inflammation. uNGAL/Cr levels correlated with creatine, highlighting the complex interactions between metabolic disruptions and tubular inflammation contributing to diabetic kidney injury.

## 4. Discussion

Our study demonstrates significantly elevated levels of tubular injury biomarkers (uMCP-1/Cr, uKIM-1/Cr, and uNGAL/Cr) in T1D patients, particularly those with albuminuria, compared to non-diabetic subjects, regardless of disease duration. These findings are consistent with prior research suggesting that these biomarkers may serve as sensitive indicators of early kidney dysfunction in T1D, preceding the onset of albuminuria [19,21,25]. In addition, we identified distinct urinary metabolic profiles associated with the progression of childhood-onset T1D. Specifically, metabolites such as NAC, NAA, NAO, and pyruvate exhibited significant differential expression and dynamic alterations across short- to long-duration T1D, independently of renal function. These metabolic signatures hold potential as biomarkers for monitoring disease progression.

In our study, pathway analysis revealed crucial alterations in amino acid and carbohydrate metabolism associated with T1D. Notably, pyruvic acid emerged as a central metabolite within branched-chain amino acid pathways, linking it to β-cell function, insulin resistance, diabetes, metabolic syndrome, and obesity [26,27,28]. Beyond its role in carbohydrate metabolism, emerging evidence suggests that pyruvate contributes to the pathophysiology of DKD. Under hyperglycemia, impaired glycolysis, mitochondrial dysfunction, oxidative stress, and activation of collateral glucose pathways (e.g., the polyol pathway) promote tubular injury and inflammation. Pyruvate may counteract these processes by sustaining glycolysis–TCA cycle flux, preserving ATP production, and scavenging reactive oxygen species, thereby limiting oxidative stress and suppressing poly (ADP-ribose) polymerase (PARP) activation [29]. In db/db (diabetic) mice, exogenous pyruvate supplementation restored the activity of metabolic enzymes (pyruvate kinase and pyruvate dehydrogenase), improved the NAD+/NADH balance, and reduced oxidative stress, advanced glycation end-product (AGE) accumulation, and mesangial expansion, thereby delaying nephropathy progression [30]. These findings suggest that pyruvate may modulate DKD development through mitochondrial protection, redox regulation, and inhibition of maladaptive glucose pathways.

Metabolites NAC, NAA, and NAO play critical roles in oxidative stress defense, energy homeostasis, and amino acid metabolism, with emerging implications in DKD progression [28,31,32]. NAC serves as a precursor to glutathione and exerts potent antioxidant effects by replenishing intracellular cysteine and enhancing enzymatic defenses such as catalase and glutathione peroxidase [33]. Experimental studies have shown that early NAC administration in diabetic rats reduces proteinuria, preserves creatinine clearance, and attenuates tubular injury, primarily through the restoration of nitric oxide bioavailability, suppression of iNOS activity, and reduction in lipid peroxidation [31]. Although NAC exhibits renoprotective properties by mitigating oxidative stress, chronic or excessive exposure may disrupt physiological ROS signaling, potentially impairing mitochondrial biogenesis and leading to reduced muscle performance or atrophy during long-term use [34]. Therefore, while elevated urinary NAC may reflect an adaptive antioxidative response in T1D, its sustained systemic increase warrants cautious interpretation.

Traditionally, NAA is considered a neuronal metabolite and as a biomarker of energy metabolism and mitochondrial function. In diabetes, reduced NAA has been linked to neuropathy and cognitive decline [35]. In addition, metabolomic studies suggest that dysregulated amino acid metabolism, including changes in NAA, may accompany DKD progression, possibly reflecting impaired renal mitochondrial integrity and energy metabolism [28]. NAO participates in the urea cycle and amino acid catabolism, functioning as an intermediate in arginine biosynthesis. A Mendelian randomization study implicates NAO-related pathways in DKD, suggesting a causal link between gut microbiota–derived metabolites and renal injury [36]. NAO accumulation may exacerbate redox imbalance, oxidative stress, and contribute to mesangial expansion and interstitial fibrosis, thus promoting DKD progression [32].

The elevated urinary levels of MCP-1/Cr, KIM-1/Cr, and NGAL/Cr observed in our cohort reflect distinct but interconnected mechanisms contributing to the pathophysiology of DKD. MCP-1, a chemokine upregulated by hyperglycemia, advanced glycation end-products, and oxidative stress, recruits monocytes/macrophages to renal tissue and drives tubulointerstitial inflammation and fibrosis through CCR2-mediated signaling, thereby contributing to albuminuria, glomerulosclerosis, and functional decline [37,38].

KIM-1, normally absent in healthy kidneys, is strongly induced in proximal tubular epithelial cells after injury. In human DKD and diabetic models, KIM-1 also mediates endocytic uptake of palmitate-bound albumin, triggering mitochondrial dysfunction, DNA damage responses, inflammation, and interstitial fibrosis [20]. Emerging evidence shows that in youth with T1D at high risk of DKD, glycemic variability is associated with higher circulating T-cell KIM-1 levels. KIM-1+T cells infiltrate DKD kidneys, and plasma from high-risk individuals induces profibrotic changes in proximal tubular cells that can be attenuated by KIM-1 blockade [39]. NGAL, secreted in response to oxidative stress and iron dysregulation, serves as a sensitive marker of early tubular stress but may also contribute to injury through altered iron transport and amplification of oxidative damage [40].

In addition, correlations between urinary metabolites (PAGly, NAC, creatine, pyruvic acid) and these three tubular injury biomarkers highlight complex metabolic responses to renal stress. PAGly, a metabolite of microbial phenylalanine metabolism, suppresses cardiomyocyte apoptosis and is linked to cardiovascular and cerebrovascular disease [41]. The association of NAC with uKIM-1/Cr underscores the role of oxidative stress in diabetic kidney injury, while correlations of pyruvic acid with uKIM-1/Cr and uMCP-1/Cr point to altered carbohydrate metabolism that exacerbates renal inflammation.

Together, these findings suggest that tubular injury biomarkers and metabolite signatures capture complementary aspects of DKD pathogenesis, linking inflammation, oxidative stress, and disordered energy metabolism to early renal injury.

## 5. Conclusions

Our findings highlight intricate interactions between metabolic disturbances and inflammation in DKD associated with T1D. Urinary biomarkers (uKIM-1/Cr, uMCP-1/Cr, uNGAL/Cr) together with specific metabolites show promise for early identification of subclinical kidney dysfunction, possibly preceding the onset of albuminuria. These characteristic metabolic signatures, particularly those linked to branched-chain amino acid metabolism, may provide valuable insights into the underlying pathophysiology of disease progression; however, given the cross-sectional design and the lack of perfectly age- and BMI-matched controls, causal relationships cannot be established. Notably, prior pediatric metabolomics studies have shown that age-related metabolic variations are modest compared with disease-related alterations in diabetes and kidney dysfunction [42,43]. Future longitudinal studies with larger, well-matched cohorts are warranted to validate these observations and to clarify whether these identified metabolites directly contribute to, or arise from, T1D progression. Stratifying participants by clinical characteristics or disease stage will further enhance the robustness and translational relevance of future research.

## Figures and Tables

**Figure 1 metabolites-15-00734-f001:**
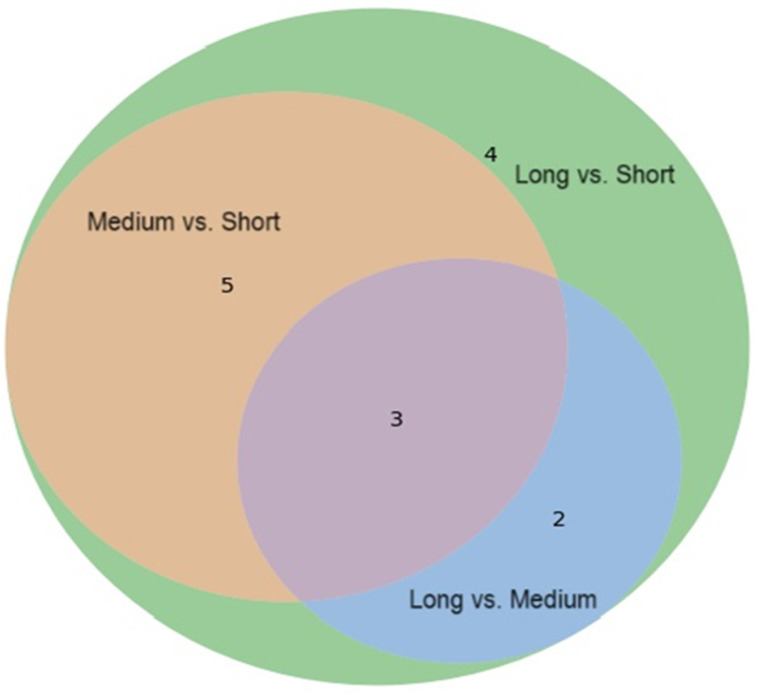
The Venn diagram illustrates the overlap of differentially expressed urinary metabolites among three comparisons: Medium vs. Short, Long vs. Short, and Long vs. Medium. Three urine metabolites (Acetylcysteine, N-acetylaspartic acid, and N-acetylornithine) are significantly differentially expressed across all three comparisons.

**Figure 2 metabolites-15-00734-f002:**
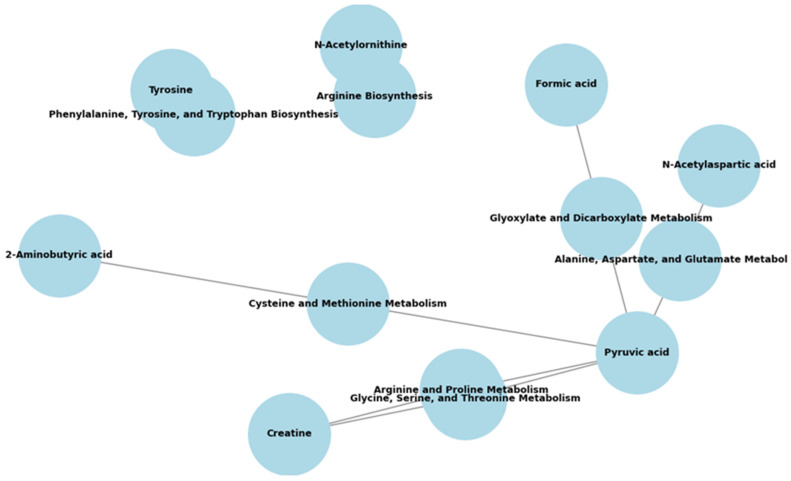
The pathway diagram illustrates how different metabolites, particularly pyruvic acid, are involved in multiple metabolic pathways. This highlights the interconnected nature of metabolic processes over time in T1D.

**Figure 3 metabolites-15-00734-f003:**
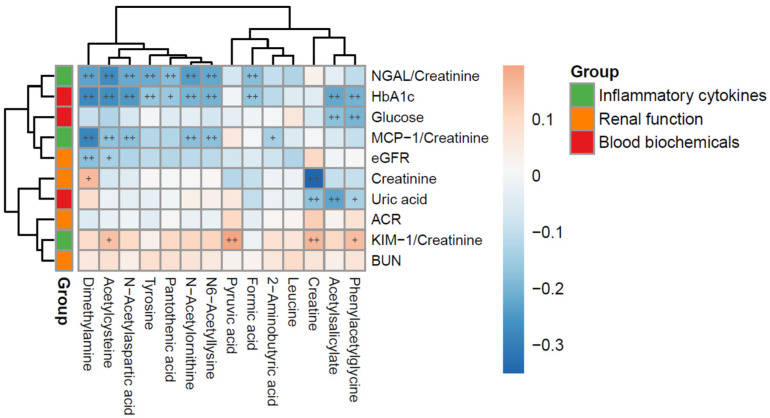
Spearman correlation heatmap of characteristic urinary metabolites, tubular inflammatory cytokines (urinary NGAL/Creatinine, MCP-1/Creatinine, and KIM-1/Creatinine), renal function markers (eGFR, urine ACR, BUN, serum creatinine), and blood biochemicals (HbA1c, glucose, uric acid) in 247 patients with childhood-onset T1D. Blue and red colors indicate negative and positive correlations, respectively. “+” denotes *p* < 0.05; “++” denotes *p* < 0.01.

**Table 1 metabolites-15-00734-t001:** Demographics and clinical characteristics for the overall cohort and the three T1D groups of short (T1D-S), medium (T1D-M), and long (T1D-L) disease durations.

		T1D Duration (Years)			Significance
	All	T1D-S (≤5)	T1D-M (6–10)	T1D-L (≥10)	
Variables	n = 247	n = 62	n = 67	n = 118	
**Demographic parameters**					
Diabetes duration (years)	9.7 (5.0, 14.0)	2.8 (1.5, 3.7)	7.6 (6.6, 8.9)	14.1 (12.6, 17.3)	0.001
Age (years)	18.6 (13.6, 22.3)	12.5 (9.3, 16.2)	16.9 (13.2, 20.6)	22.0 (19.0, 25.9)	0.001
Age at diagnosis (years)	8.2 (4.7, 11.7)	9.8 (6.3, 13.4)	9.8 (5.9, 12.4)	6.7 (4.0, 10.2) *	0.001
Male/female (number, %)	121 (49.0%)/126 (51.0%)	31 (44.9%)/38 (55.1%)	35 (53.0%)/31 (47.0%)	55 (49.1%)/57 (50.9%)	
BMI (kg/m^2^)	21.0 (18.6, 23.0)	18.7 (16.0, 20.4)	20.9 (17.9, 23.2)	22.1 (20.5, 24.4)	0.001
SBP (mmHg)	118 (106, 128)	108 (102, 118)	118 (104, 128)	124 (113, 135)	0.001
DBP (mmHg)	71 (64, 79)	66 (60, 72)	70 (64, 77)	76 (67, 82)	0.001
**Glycemic control**					
FBS (mg/dL)	166 (109, 243)	150 (100, 214)	184 (99, 246)	167 (120, 243)	0.61
HbA1c (%)	8.2 (7.3, 9.4)	8.2 (7.1, 9.1)	8.0 (7.3, 9.4)	8.4 (7.4, 9.5)	0.70
**Plasma parameters**					
TC (mg/dL)	176 (158, 202)	172 (153, 198)	173 (152, 210)	178 (164, 203)	0.16
LDL-C (mg/dL)	99 (81, 120)	90 (78, 118)	94 (81, 123)	104 (85, 119) *	0.05
HDL-C (mg/dL)	61 (53, 72)	62 (51, 73)	62 (54, 73)	61 (54, 70)	0.78
Triglyceride (mg/dL)	59 (43, 84)	59 (40, 79)	55 (42, 90)	63 (48, 85)	0.70
Uric acid (mg/dL)	4.5 (3.7, 5.3)	4.1 (3.2, 5.0)	4.5 (3.6, 5.4)	4.6 (4.0, 5.3) *	0.02
hs-CRP (mg/dL)	0.6 (0.2, 1.6)	0.3 (0.2, 0.87)	0.7 (0.2, 1.51)	0.8 (0.22, 2.06)	0.41
Homocysteine (μmol/L)	7.9 (6.6, 9.6)	6.9 (6.1, 8.9)	7.7 (6.4, 9.3)	8.2 (7.2, 10.4) *	0.01
**Hematological parameters**					
WBC (1000/uL)	6.2 (5.2, 7.5)	6.0 (5.0, 7.0)	6.2 (5.3, 7.6)	6.2 (5.2, 7.6)	0.58
Hb (g/dL)	14.0 (13.2, 15)	13.7 (13.2, 14.2) *	14.0 (13.1, 14.8)	14.5 (13.3, 15.3)	0.01
Platelet (1000/uL)	283 (240, 327)	279.0 (232.0, 317.0)	275.0 (231.0, 329.0)	289.5 (246.0, 328.0)	0.34
**Nephrology**					
Urine ACR (mg/g)	5.6 (3.6, 10.6)	6.3 (4.0, 10.6)	5.4 (3.6, 9.5)	5.4 (3.6, 12.4)	0.73
eGFR (ml/min/1.73 m^2^) *	133.4 (114.2, 155.8)	132.2 (116.9, 144.3)	131.3 (110.9, 160.8)	137.2 (115.1, 159.4)	0.34

BMI, body mass index; SBP, systolic blood pressure; DBP, diastolic blood pressure; FBS, fasting blood sugar; HbA1c, glycated hemoglobin; TC, total cholesterol. LDL-C, low-density lipoprotein cholesterol. HDL-C, high-density lipoprotein cholesterol; hs-CRP, high-sensitivity C-reactive protein. WBC, white blood cell. Hb, hemoglobulin. UACR, urine albumin-to-creatinine ratio; eGFR, estimated glomerular filtration rate. Data expressed as median (IQR). Comparisons among the T1D subgroups were analyzed using Kruskal–Wallis ANOVA, followed by post hoc Tukey tests (OriginPro (2023) software). * A *p*-value < 0.05 was considered statistically significant.

**Table 2 metabolites-15-00734-t002:** Comparative analysis of urinary cytokines, MCP-1, KIM-1, and NGAL, in T1D-S (0–5 years), T1D-M (6–10 years), and T1D-L (>10 years) patients versus non-diabetic subjects.

			T1D Patients		
	Non-Diabetes (n = 60)	All T1D (n = 247)	T1D-S (n = 62)	T1D-M (n = 67)	T1D-L (n = 118)
Age (years)	9.2 (8.4, 10.8)	18.6 (13.6, 22.3) *	12.5 (9.3, 16.4)	16.9 (13.2, 20.6) *	22.0 (18.9, 25.6) *
Male (%)	24 (40.0%)	121 (49.0%)	31 (44.9%)	35 (53.0%)	55 (49.1%)
BMI (kg/m^2^)	17.2 (15.4, 19.5)	21.0 (18.6, 23.0) *	18.7 (16.0, 20.4)	20.9 (17.9, 23.2) *	22.1 (20.5, 24.4) *
Urine cytokines					
uMCP-1/Cr (µg/g)	0 (0, 0.05)	0.11 (0.07, 0.20) *	0.14 (0.08, 0.25) *	0.14 (0.07, 0.22) *	0.10 (0.06, 0.17) *
uKIM-1/Cr (µg/g)	0.39 (0.20, 0.61)	0.68 (0.37, 1.21) *	0.85 (0.41, 1.51) *	0.85 (0.46, 1.49) *	0.56 (0.30, 1.03) *
uNGAL/Cr (µg/g)	2.53 (1.78, 4.22)	7.56 (3.44, 25.93) *	7.45 (4.68, 23.57) *	7.38 (3.47, 18.21) *	7.96 (2.93, 31.24) *

uMCP-1, urine monocyte chemoattractant protein-1; uKIM-1, urine kidney injury molecule-1; uNGAL, urine neutrophil gelatinase-associated lipocalin. Data is presented as median (IQR) or as a number (%). * Significance is at *p* value < 0.05 for comparison of groups by age, BMI, uMCP-1, uKIM-1, and uNGAL with Kruskal–Wallis ANOVA and post hoc analysis (OriginPro (2023) software).

**Table 3 metabolites-15-00734-t003:** Binary logistic regression analyses of urinary cytokines (uMCP-1/Cr, uKIM-1/Cr, and uNGAL/Cr) and covariates associated with albuminuria in T1D patients.

Urine Cytokines (µg/g)	UACR < 30 mg/g (n = 31) ^ф^	UACR ≧ 30 mg/g (n = 216) ^ф^	*p*-Value *	Adjusted Odds Ratio ^Ψ^ (95% CI)	*p*-Value *
uMCP-1/Cr	0.108 (0.068, 0.189)	0.177 * (0.071, 0.261)	0.031	0.85 (0.09–7.80)	0.887
uKIM-1/Cr	0.664 (0.364, 1.188)	0.962 * (0.504, 2.488)	0.020	0.74 (0.46–1.18)	0.209
uNGAL/Cr	6.802 (3.028, 21.492)	17.123 * (7.790, 55.674)	0.0007	0.99 (0.98–1.00)	0.078

^ф^ Values are presented as medians (IQR) and analyzed using Kruskal–Wallis ANOVA for group comparisons. ^Ψ^ Binary logistic regression models adjusted for age, sex, BMI, HbA1c, eGFR, and diabetes duration. * Statistical significance was defined as *p* < 0.05 (OriginPro (2023) software).

**Table 4 metabolites-15-00734-t004:** The VIP score and fold change in urine metabolites significantly differentially expressed between T1D patients with short-duration, medium-duration, and long-duration follow-ups.

		Medium vs. Short			Long vs. Short			Long vs. Medium		
Metabolites	Chemical Shift	VIP Score *	Fold Change †	*p* ‡	VIP Score	Fold Change	*p*	VIP Score	Fold Change	*p*
Acetylcysteine	2.081–2.092 (s)	1.58	0.73	**0.001**	2.53	0.46	**<0.001**	2.10	0.63	**<0.001**
N-Acetylaspartic acid	2.026–2.030 (s)	1.40	0.77	**0.001**	1.67	0.65	**<0.001**	1.04	0.85	**0.009**
N-Acetylornithine	2.045–2.053 (s)	0.91	0.83	0.026	1.20	0.73	**<0.001**	0.83	0.88	0.031
Acetylsalicylate	2.340–2.355 (s)	1.87	0.65	**0.002**	2.08	0.52	**<0.001**	1.06	0.80	0.067
Phenylacetylglycine	7.423–7.446 (m)	1.40	0.75	**0.006**	1.58	0.63	**<0.001**	0.79	0.83	0.101
Dimethylamine	2.718–2.734 (s)	0.93	0.86	**0.006**	0.64	1.03	0.016	0.57	1.20	0.854
Creatine	3.933–3.943 (s)	2.96	0.35	0.025	1.83	0.33	0.042	1.14	0.94	0.564
2-Aminobutyric acid	0.982–0.987 (t)	1.57	0.65	0.033	0.95	0.71	**0.005**	1.07	1.09	0.221
Pyruvic acid	2.379–2.388 (s)	0.81	0.90	0.123	1.54	0.75	**<0.001**	1.60	0.83	**0.008**
N6-Acetyllysine	1.987–1.994 (s)	0.74	0.85	0.063	1.05	0.76	**<0.001**	0.76	0.89	0.026
Formic acid	8.460–8.470 (s)	0.92	0.80	0.078	1.19	0.66	**0.001**	0.88	0.83	0.123
Tyrosine	6.891–6.916 (m)	0.74	0.83	0.123	0.80	0.75	**0.008**	0.40	0.89	0.449
Pantothenic acid	0.934–0.940 (s)	0.81	0.82	0.064	0.77	0.80	**0.009**	0.66	0.97	0.508
Leucine	0.951–0.968 (t)	0.87	0.74	0.083	0.81	0.68	**0.014**	0.37	0.93	0.624

Data are reported as chemical shift ranges (δ, ppm). * VIP scores were obtained from partial least-squares discriminant analysis (PLS-DA). † Fold change was calculated by dividing metabolite levels between T1D patients with short-, medium-, and long-term duration. ‡ Only metabolites that met both the conservative significance threshold (*p* < 0.01) and false discovery rate (FDR) correction (q < 0.05) were considered statistically significant and are shown in bold. Abbreviations: VIP, variable importance in projection; ppm, parts per million; m, multiplet; s, singlet; t, triplet.

**Table 5 metabolites-15-00734-t005:** Metabolic pathway and functional analysis of urinary metabolites identified in T1D patients with varying disease durations.

Metabolites	Pathway Name	Total	Hits	Raw *p*	FDR	Function
**T1D-M vs. T1D-S**						
None						
**T1D-L vs. T1D-S**						
N-Acetylaspartic acid, pyruvic acid	Alanine, aspartate, and glutamate metabolism	28	2	0.025	0.299	Amino acid metabolism
Pyruvic acid, formic acid	Glyoxylate and dicarboxylate metabolism	32	2	0.032	0.599	Carbohydrate metabolism
Creatine, pyruvic acid	Glycine, serine, and threonine metabolism	33	2	0.034	0.299	Amino acid metabolism
2-Aminobutyric acid, pyruvic acid	Cysteine and methionine metabolism	33	2	0.034	0.299	Amino metabolism
Tyrosine	Phenylalanine, tyrosine, and tryptophan biosynthesis	4	1	0.036	0.599	Amino acid metabolism
Creatine, pyruvic acid	Arginine and proline metabolism	38	2	0.044	0.619	Amino acid metabolism
**T1D-L vs. T1D-M**						
N-Acetylaspartic acid, pyruvic acid	Alanine, aspartate, and glutamate metabolism	28	2	0.003	0.256	Amino acid metabolism
N-Acetylornithine	Arginine biosynthesis	14	1	0.044	1.000	Amino acid metabolism

Total is the total number of compounds in the pathway; Hits is the actually matched number from the user uploaded data; Raw *p* is the original *p* value calculated from the enrichment analysis; FDR is the portion of false positives above the user-specified score threshold. (FDR, false discovery rate).

## Data Availability

The metabolomics dataset generated and analyzed in this study has been deposited in the NIH Metabolomics Workbench public repository (DataTrack ID: 6504; Study ID: ST004325; DOI: http://dx.doi.org/10.21228/M8MG1P) under the title “Dynamics of urine metabolomics and tubular inflammatory cytokines in Type 1 diabetes across disease durations.” Processed data and statistical analysis results are available from the corresponding author upon reasonable request, in accordance with institutional ethics and IRB regulations.

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
