# Peer review of "Dynamics of Urine Metabolomics and Tubular Inflammatory Cytokines in Type 1 Diabetes Across Disease Durations"

_metabolites, 2025, doi:10.3390/metabo15110734_

Round 1

Reviewer 1 Report (Previous Reviewer 2)

Comments and Suggestions for Authors

All points raised in the first version of the manuscript have been appropriately addressed by the authors.

The manuscript is of publication quality

Author Response

Comments 1: All points raised in the first version of the manuscript have been appropriately addressed by the authors.   2.The manuscript is of publication quality

Point 1:  The English is fine and does not require any improvement.

Reviewer 2 Report (Previous Reviewer 1)

Comments and Suggestions for Authors

I am generally satisfied with the authors’ revisions and responses and have no further comments.

Author Response

Comment: I am generally satisfied with the authors’ revisions and responses and have no further comments.

Reviewer 3 Report (Previous Reviewer 3)

Comments and Suggestions for Authors

This manuscript presents useful cytokine and metabolomics datasets and data analysis that provide useful insight into biomarkers associated with type 1 diabetes and diabetic kidney disease.  The data analyses are mostly rigorously performed and the results are generally well presented.  However, there are still major issues that must be addressed. 

Very Major Issues:
A) The datasets are not deposited in a public scientific repository to support both scientific reproducibility and reusability of these valuable datasets.  Furthermore, the “Data Availability Statement” is very misleading, since only aggregate statistics are provided in supplemental material.  Given that these datasets are derived from human patients, the datasets must be deidentified.  And there is the possibility that the IRB used for the sample collection precludes public data deposition.  However, this reason has NOT been provided.  In 21st century science, is COMPLETELY UNACCEPTABLE not to make your datasets publicly available to support scientific reproducibility.

Major Issues:
1. No justification is provided for the age-specific grouping of patients in the Materials and Methods section (around line 92).  The following is one paper that can help in the justification:
Baek JH, Lee WJ, Lee BW, Kim SK, Kim G, Jin SM, Kim JH. Age at diagnosis and the risk of diabetic nephropathy in young patients with type 1 diabetes mellitus. Diabetes & metabolism journal. 2021 Jan 1;45(1):46-54.
3. Are the given metabolite vs disease duration correlation results potentially confounded with patient age?  Was this investigated?   The authors should either demonstrate they are not potentially confounded or discuss this possibility.  Even if they are confounded, this confound can be ruled out by including a metabolite vs age correlation analysis of healthy controls that span the same age ranges as the disease groups and demonstrating a lack of correlation.  There should be public datasets available for this purpose, but this reviewer is NOT requiring such an analysis in this manuscript. There might be a paper that has done such an analysis that can be referenced instead.
3. The authors cite the positive aspects of N-acetyl-cysteine with respect to diabetic kidney disease, but the authors should balance with references to possible long-term side effects of elevated N-acetylc-cysteine, for example with respect to reduced muscle performance and atrophy.

Minor Issues:
Line 153: Should mention and cite the specific Python packages used for generating Venn diagrams and pathway diagrams.

Author Response

Comment 1: The datasets are not deposited in a public scientific repository to support both scientific reproducibility and reusability of these valuable datasets. Furthermore, the “Data Availability Statement” is very misleading, since only aggregate statistics are provided in supplemental material.  Given that these datasets are derived from human patients, the datasets must be deidentified.  And there is the possibility that the IRB used for the sample collection precludes public data deposition.  However, this reason has NOT been provided.  In 21st century science, is COMPLETELY UNACCEPTABLE not to make your datasets publicly available to support scientific reproducibility.

Response :We have already submitted our metabolomics dataset to the Metabolomics Workbench public repository. According to the Metabolomics Workbench guidelines, “It typically takes 5–10 working days for a submission to be reviewed and processed. The submitter will then be notified and provided with a DOI and a private link to the study which may be shared with reviewers.”

As the review process is still ongoing, we would like to provide the following materials as supporting evidence of our data deposition:

  • A screenshot of the Metabolomics Workbench submission page.
  • The consolidated mwTab file, which contains all uploaded metadata and processed data.

We will promptly provide the DOI and study link as soon as the submission has been reviewed and approved by the Metabolomics Workbench team.

2. Comment 2: No justification is provided for the age-specific grouping of patients in the Materials and Methods section (around line 92). The following is one paper that can help in the justification:

Baek JH, Lee WJ, Lee BW, Kim SK, Kim G, Jin SM, Kim JH. Age at diagnosis and the risk of diabetic nephropathy in young patients with type 1 diabetes mellitus. Diabetes & metabolism journal. 2021 Jan 1;45(1):46-54.

Response:

   We thank the reviewer for this valuable comment. In our study, the primary objective was to investigate the impact of disease duration on metabolic and inflammatory changes in T1D. Therefore, our grouping (short ≤ 5 years; medium 6–10 years; long > 10 years) was based on duration since diagnosis, rather than age at onset.

   Nevertheless, we agree that age at diagnosis is a clinically important modifier of diabetic complications. The study by Baek et al. (2021, Diabetes & Metabolism Journal, 45:46–54) demonstrated that younger age at T1D onset (< 20 years) was independently associated with a higher risk of macroalbuminuria and diabetic nephropathy compared with onset during early adulthood (20–40 years). Their work provides valuable context supporting age-related heterogeneity in kidney outcomes.

   In our cohort, all participants had childhood-onset T1D, with a median age at diagnosis of approximately 8 years (IQR 4.7–11.7)—substantially younger than the adult-onset groups analyzed in Baek et al. Because the entire cohort fell within the youth-onset category, further stratification by age at diagnosis (e.g., < 20 years vs. ≥ 20 years) was not applicable. Nevertheless, we fully agree that future longitudinal work could further examine whether metabolic and inflammatory signatures differ between pre-pubertal and post-pubertal onset of T1D, given evidence that puberty may accelerate diabetic kidney injury.

Revision:

  • Line 100: Clinical parameters, including age, age at diagnosis, gender, body mass index (BMI)…..
  • Line 177: As anticipated, the age of diagnosis, disease duration, blood pressure (SBP, DBP) and BMI increased with T1D duration, and were higher in the T1D-L group, reflecting their older age.
  • Table 1 (revised): “ Age” and “ Age at diagnosis"

3. Comment 3:  Are the given metabolite vs disease duration correlation results potentially confounded with patient age?  Was this investigated?   The authors should either demonstrate they are not potentially confounded or discuss this posTsibility.  Even if they are confounded, this confound can be ruled out by including a metabolite vs age correlation analysis of healthy controls that span the same age ranges as the disease groups and demonstrating a lack of correlation.  There should be public datasets available for this purpose, but this reviewer is NOT requiring such an analysis in this manuscript. There might be a paper that has done such an analysis that can be referenced instead.  

Response:

   We sincerely thank the reviewer for this insightful comment. In our study, we carefully addressed the potential confounding between patient age and metabolite–disease duration correlations. Specifically, age was included as a covariate in all multivariate and correlation analyses (see Supplementary Table S3), ensuring that observed associations between urinary metabolites and T1D duration were adjusted for age-related variation.

   Although our cohort spans a relatively narrow pediatric–adolescent range, prior metabolomics literature indicates that age-related differences in urinary metabolites among healthy children are modest compared with the disease-related metabolic alterations observed in diabetes. For example:

  • Bervoets et al. (2017, Diabetology & Metabolic Syndrome, 9:48) similarly found that the metabolic distinctions between T1D patients and controls greatly exceeded intra-age variation, confirming that disease status rather than chronological age primarily drives metabolic divergence.
  • Hendrix et al. (2023, Endocrinol Diabetes Metab. 2023, 6, e448 ) demonstrated that alterations in branched-chain amino acid catabolism and fatty acid metabolism in children with T1D occur independently of age, supporting our interpretation that these metabolic differences reflect disease-related changes rather than maturational effects.

These consistent findings suggest that while age may influence certain baseline metabolites (e.g., creatinine or citrate), it is unlikely to explain the significant metabolic differences linked to T1D duration observed in our cohort.

Revisions:

  1. Line 480-482: Notably, prior pediatric metabolomics studies have shown that age-related metabolic variations are modest compared with disease-related alterations in diabetes and kidney dysfunction [41, 42].
  2. Newly added references [42] and [43]:                                                                                                                                  42. Bervoets, L.; Massa, G.; Guedens, W.; Louis, E.; Noben, J.P.; Adriaensens, P. Metabolic profiling of type 1 diabetes mellitus in children and adolescents: a case-control study. Diabetol Metab Syndr. 2017, 9, 48, doi:10.1186/s13098-017-0246-9.                                                                                                                                                                                43.Hendrix, G.; Lokhnygina, Y.; Ramaker, M.; Ilkayeva, O.; Muehlbauer, M.; Evans, W.; Rasbach, L.; Benjamin, R.; Freemark, M.; Gumus Balikcioglu, P. Catabolism of fats and branched-chain amino acids in children with Type 1 diabetes: Association with glycaemic control and total daily insulin dose. Endocrinol Diabetes Metab. 2023, 6, e448, doi:10.1002/edm2.448.

4. Comment 4 : The authors cite the positive aspects of N-acetyl-cysteine with respect to diabetic kidney disease, but the authors should balance with references to possible long-term side effects of elevated N-acetylc-cysteine, for example with respect to reduced muscle performance and atrophy.  

Response:

Thank you for this valuable suggestion. We have made the following revision:

  • Lines 434–437 (Revised): Although NAC exhibits renoprotective properties by mitigating oxidative stress, chronic or excessive exposure may disrupt physiological ROS signaling, potentially impairing mitochondrial biogenesis and leading to reduced muscle performance or at-rophy during long-term use [34].
  • Newly added reference [34]: Merry, T.L.; Ristow, M. Do antioxidant supplements interfere with skeletal muscle adaptation to exercise training? J Physiol 2016, 594, 5135-5147, doi:10.1113/jp270654.

5. Comment 5: Line 153: Should mention and cite the specific Python packages used for generating Venn diagrams and pathway diagrams.

Response:

Thank you for this helpful comment. We have revised the Materials and Methods (Section 2.4, around Line 153-155) to specify the Python packages used for visualization.

The revised text: Venn diagrams were generated using the matplotlib-venn and matplotlib packages in Python 3, while pathway diagrams were created using networkx and matplotlib. 

Round 2

Reviewer 3 Report (Previous Reviewer 3)

Comments and Suggestions for Authors

The authors have well-addressed the issues raised by this reviewer.

However, it would be prudent to wait for the Metabolomics Workbench deposition to process so that the DOI for the dataset can be added to the Data Availability Statement.  One possible alternative, is to state that the dataset has been submitted to Metabolomics Workbench on a given date under the submitted title and then provide the date and title.

Author Response

Comment: However, it would be prudent to wait for the Metabolomics Workbench deposition to process so that the DOI for the dataset can be added to the Data Availability Statement.  One possible alternative, is to state that the dataset has been submitted to Metabolomics Workbench on a given date under the submitted title and then provide the date and title.

Response: 

Thank you for your valuable suggestion. We have deposited the metabolomics dataset to the Metabolomics Workbench public database (DataTrack ID: 6504) under the title “Dynamics of urine metabolomics and tubular inflammatory cytokines in Type 1 diabetes across disease durations.”

The dataset was initially submitted on October 23, 2025, and reconfirmed on October 27, 2025 to ensure proper registration.

According to the Metabolomics Workbench guideline, “It typically takes 5–10 working days for a submission to be reviewed and processed. The submitter will then be notified and provided with a DOI and a private link to the study which may be shared with reviewers.”

The uploaded dataset information is available at:

https://www.metabolomicsworkbench.org/data/show_mwtabfile.php?F=msyu_0403_20250929_042843_mwtab.txt

Once the curation process is completed and a DOI is assigned, we will update the Data Availability Statement accordingly.

Revision:  Data Availability Statement (Line 517-521)

The metabolomics dataset generated and analyzed in this study has been deposited in the    Metabolomics Workbench public repository under DataTrack ID: 6504, titled “Dynamics of urine metabolomics and tubular inflammatory cytokines in Type 1 diabetes across disease durations.” Processed data and corresponding statistical analysis results are available from the corresponding author upon reasonable request, in accordance with institutional ethics and IRB regulations.

This manuscript is a resubmission of an earlier submission. The following is a list of the peer review reports and author responses from that submission.

Round 1

Reviewer 1 Report

Comments and Suggestions for Authors

The manuscript presents a cross-sectional study investigating urinary tubular injury biomarkers and metabolomic profiles in youth-onset Type 1 diabetes (T1D) patients across different disease durations. The study is well-designed, with a robust sample size (n=247) and a comprehensive approach integrating urinary cytokines (MCP-1, KIM-1, NGAL) and metabolomic analysis using 1H-NMR spectroscopy. The findings, particularly the identification of characteristic metabolites (NAC, NAA, NAO, pyruvic acid) and their association with diabetic kidney disease (DKD), are novel and contribute to the field. However, several areas require significant improvement to enhance clarity, scientific rigor, and the manuscript's overall impact. I recommend major revisions before the manuscript can be considered for publication.

Major Issues

1.Discrepancies in Sample Size Reporting:

The abstract (line 29) states the sample sizes as T1D-S (n=62), T1D-M (n=67), and T1D-L (n=118), but the methods section (line 88) reports T1D-S (n=69), T1D-M (n=66), and T1D-L (n=112). This inconsistency is critical and must be resolved. Please clarify the correct sample sizes and ensure consistency throughout the manuscript, including tables and figures.

Additionally, the correlation heatmap (Figure 3, line 352) mentions n=217, which does not align with the total cohort of n=247. Explain the reason for this discrepancy (e.g., missing data, exclusion criteria) and address it consistently in the text and figures.

2.Incomplete Description of Control Group:

The study compares T1D patients to non-diabetic controls (Table 2, line 210), but the methods section lacks details about the control group (e.g., recruitment criteria, sample size, demographic characteristics, and how they were matched to T1D patients). This is critical for interpreting the significance of elevated urinary cytokine levels in T1D patients. Please provide a detailed description of the control group in the methods section.

3.Statistical Analysis Clarity:

The statistical methods (Section 2.5, lines 130–134) describe the use of Kruskal-Wallis ANOVA, Dunn’s tests, and logistic regression but lack detail on adjustments for multiple comparisons, particularly for metabolomic analyses. Given the large number of metabolites tested, the risk of false positives is high. Clarify whether corrections (e.g., Bonferroni or FDR) were applied beyond the mentioned FDR in Table 5 (line 246).

The logistic regression results (Table 3, line 225) report coefficients and odds ratios for urinary cytokines but do not explain the model’s covariates or adjustments (e.g., age, sex, HbA1c). Provide a detailed description of the regression model, including all covariates and their rationale.

4.Metabolite Interpretation and Biological Relevance:

The discussion (Section 4, lines 357–398) provides a good overview of the roles of NAC, NAA, NAO, and pyruvic acid but lacks depth in linking these metabolites to specific pathophysiological mechanisms in T1D and DKD. For example, the role of pyruvic acid in carbohydrate metabolism is mentioned, but its specific contribution to renal inflammation or tubular injury is unclear. Expand the discussion to include mechanistic insights, supported by additional references if necessary.

The manuscript claims that NAC, NAA, and NAO are “closely linked to T1D progression” (line 42), but the cross-sectional design limits causal inferences. Revise such statements to reflect the study’s observational nature and suggest longitudinal studies to confirm causality.

5.Figure and Table Presentation:

Table 1 (line 201): The table is comprehensive but difficult to read due to dense formatting. Consider splitting it into multiple tables (e.g., demographic, glycemic, lipid, and renal parameters) to improve readability. Additionally, clarify the meaning of “NS” (not significant) and provide exact p-values where possible.

  1. Methodological Details for Metabolomics:

The metabolomic analysis (Section 2.3, lines 109–120) describes the use of 1H-NMR spectroscopy and MetaboAnalyst but lacks specifics on quality control measures (e.g., batch effects, normalization procedures) and the rationale for selecting specific metabolites (e.g., VIP score ≥ 1.0). Provide a more detailed description of the metabolomic pipeline, including quality control steps and criteria for metabolite selection.

The use of both intelligent and variable size bucketing (line 117) is mentioned but not justified. Explain why both techniques were used and how they impacted the results.

Minor Issues

1.Terminology Consistency:

The manuscript uses “uMCP-1,” “uKIM-1,” and “uNGAL” interchangeably with “MCP-1/Cr,” “KIM-1/Cr,” and “NGAL/Cr.” Standardize terminology (e.g., consistently use “uMCP-1/Cr” for creatinine-normalized values) to avoid confusion.

2.Define abbreviations (e.g., PLS-DA, VIP) the first time they are used in the text, not just in the abbreviations section.

3.Typographical and Formatting Errors:

Line 88: The term “childhood-onset T1D” should be consistently used instead of “youth-onset” (e.g., line 138) to align with WHO criteria mentioned.

Line 428: “Informed Consent Statement)” appears incomplete and should be corrected to “Institutional Review Board” or similar.

4.Reference Formatting:

Some references (e.g., lines 440–542) are incomplete or inconsistently formatted (e.g., missing journal names or DOIs). Ensure all references follow the journal’s citation style and are complete.

5.Clarity in Results:

In Table 4 (line 236), the chemical shift ranges for metabolites (e.g., “2.081–2.092 (s)”) are provided, but the abbreviation “s” (singlet) is not explained in the table legend. Include a clear legend defining all abbreviations (e.g., s, m, t).

The results section (lines 136–188) is dense and could benefit from subheadings to separate demographic, cytokine, and metabolomic findings for better readability.

Author Response

Comments and Suggestions for Authors

The manuscript presents a cross-sectional study investigating urinary tubular injury biomarkers and metabolomic profiles in youth-onset Type 1 diabetes (T1D) patients across different disease durations. The study is well-designed, with a robust sample size (n=247) and a comprehensive approach integrating urinary cytokines (MCP-1, KIM-1, NGAL) and metabolomic analysis using 1H-NMR spectroscopy. The findings, particularly the identification of characteristic metabolites (NAC, NAA, NAO, pyruvic acid) and their association with diabetic kidney disease (DKD), are novel and contribute to the field. However, several areas require significant improvement to enhance clarity, scientific rigor, and the manuscript's overall impact. I recommend major revisions before the manuscript can be considered for publication.

Major Issues

  1. Discrepancies in Sample Size Reporting:

The abstract (line 29) states the sample sizes as T1D-S (n=62), T1D-M (n=67), and T1D-L (n=118), but the methods section (line 88) reports T1D-S (n=69), T1D-M (n=66), and T1D-L (n=112). This inconsistency is critical and must be resolved. Please clarify the correct sample sizes and ensure consistency throughout the manuscript, including tables and figures.

Additionally, the correlation heatmap (Figure 3, line 352) mentions n=217, which does not align with the total cohort of n=247. Explain the reason for this discrepancy (e.g., missing data, exclusion criteria) and address it consistently in the text and figures.

 Reply:

(1) We sincerely apologize for the inconsistency in sample size reporting. The correct sample sizes are as stated in the abstract (Line 29-31) and Table 1: T1D-S (n=62), T1D-M (n=67), and T1D-L (n=118). We have carefully revised the methodology section (Line 91-92) to reflect these numbers and ensured consistency across the entire manuscript, including all tables and figures.

(2) We also apologize for the typographical error in the correlation heatmap (Line 394). The correct sample size is n=247, consistent with the total cohort. The previously stated n=217 was an author error, and we have corrected it in the figure and corresponding text. To further confirm accuracy, we can provide the original dataset for reference upon request.

  1. Incomplete Description of Control Group:

The study compares T1D patients to non-diabetic controls (Table 2, line 210), but the methods section lacks details about the control group (e.g., recruitment criteria, sample size, demographic characteristics, and how they were matched to T1D patients). This is critical for interpreting the significance of elevated urinary cytokine levels in T1D patients. Please provide a detailed description of the control group in the methods section.

Reply:

We sincerely appreciate the reviewer’s insightful comment.

In response, we have provided a detailed description of the control group in the Methods section (Line 93–99): Furthermore, 60 subjects served as non-diabetic controls for comparison of urinary cytokine levels with T1D patients. These individuals were initially referred to the pediatric nephrology clinic for evaluation of suspected urinary abnormalities (e.g., abnormal urine test results or voiding dysfunction such as urinary frequency). Following comprehensive assessment—including repeat urinalysis, blood tests for kidney function, and renal/bladder ultrasonography—all control subjects were confirmed to have normal renal function and no structural abnormalities.

  1. Statistical Analysis Clarity:

The statistical methods (Section 2.5, lines 130–134) describe the use of Kruskal-Wallis ANOVA, Dunn’s tests, and logistic regression but lack detail on adjustments for multiple comparisons, particularly for metabolomic analyses. Given the large number of metabolites tested, the risk of false positives is high. Clarify whether corrections (e.g., Bonferroni or FDR) were applied beyond the mentioned FDR in Table 5 (line 246).

Reply:

We thank the reviewer for raising this important point.

For the metabolomic analyses, we applied multiple testing correction to minimize the risk of false positives. Specifically, pathway enrichment analyses were corrected using the False Discovery Rate (FDR), as shown in Table 5. For pairwise comparisons of individual metabolites (Tables 4), we initially applied non-parametric Mann–Whitney tests and identified metabolites with raw p < 0.05. To further validate the robustness of these findings, we incorporated a Random Forest model with Boruta feature selection and cross-validation, which consistently highlighted N-acetylcysteine, N-acetylaspartic acid, and N-acetylornithine as the most discriminative metabolites across groups. This combined approach aims to reduce the likelihood of spurious associations. Accordingly, we clarify these details in the revised Methods section (Section 2.4 and Section 2.5)

2.4. Metabolite differential analysis and pathway evaluation (Line 145-154)

Differences in metabolites between groups were assessed using Mann–Whitney tests (MetaboAnalyst). To account for multiple comparisons, p-values were adjusted using the false discovery rate (FDR) method, and adjusted q-values were reported. For pairwise comparisons of individual metabolites, features with raw p < 0.05 were initially identified, and the robustness of these findings was further validated through Random Forest modeling combined with Boruta feature selection and 20-fold cross-validation. In addition, correlations between urinary metabolites, biochemical parameters, and urinary cytokines were examined using Spearman’s correlation analysis (R software v4.1). Venn diagrams and pathway diagrams were generated using Python 3.

2.5. Statistical analyses (Line 155-169, and Supplementary Table S1 is newly added)

Descriptive statistics were expressed as medians with interquartile ranges (IQR). Normality of continuous variables was assessed using the Shapiro–Wilk test, and many variables demonstrated significant deviations from normality (Supplementary Table S1). Therefore, comparisons across subgroups and by albuminuria status were performed using Kruskal–Wallis ANOVA followed by Dunn’s post hoc tests. Binary logistic regression was applied to assess associations between urinary cytokines and albuminuria, adjusting for covariates including HbA1c, age, sex, BMI, eGFR, and      diabetes duration. Pearson correlation coefficients were also calculated. For metabolomic data, p-values from Mann–Whitney tests were adjusted for multiple testing using the Benjamini–Hochberg FDR method, with adjusted q < 0.05 considered statistically significant. In addition, to further reduce the risk of false discovery from multiple testing, statistical significance was also conservatively defined as p < 0.01. All tests were two-tailed. Analyses were conducted using IBM SPSS Statistics version 20.0 (IBM Corp., Armonk, NY, USA) and OriginPro 2023 (OriginLab Corp., Northampton, MA, USA).

The logistic regression results (Table 3, line 225) report coefficients and odds ratios for urinary cytokines but do not explain the model’s covariates or adjustments (e.g., age, sex, HbA1c). Provide a detailed description of the regression model, including all covariates and their rationale.

Reply:

In the original Table 3, binary logistic regression was performed with albuminuria (ACR <30 vs. ≥30 mg/g) as the dependent variable and individual urinary cytokines (MCP-1, KIM-1, or NGAL) as predictors, without adjustment for covariates. Following your suggestion, we reanalyzed the data using multivariable logistic regression models that included additional covariates (HbA1c, age, sex, BMI, diabetes duration, and eGFR).

The uMCP-1/Cr, uKIM-1/Cr, and uNGAL/Cr models as a whole were statistically significant: Likelihood Ratio χ²(7) = 39.06 (p < 0.0001), χ²(7) = 41.42 (p < 0.0001), and χ²(7) = 41.69 (p < 0.0001), respectively:

  • uMCP-1/Cr: Not an independent predictor of albuminuria after adjustment (OR = 0.85, 95% CI: 0.09–7.80, p = 0.887).
  • uKIM-1/Cr: Not independently associated with albuminuria after adjustment (OR = 0.74, 95% CI: 0.46–1.18, p = 0.209).
  • uNGAL/Cr: Showed a weak trend toward lower odds of albuminuria, but this did not reach statistical significance after adjustment (OR = 0.99, 95% CI: 0.98–1.00, p = 0.078).

Revision: (Line 185-209)

  • Urinary MCP-1/Cr, KIM-1/Cr, and NGAL/Cr levels in T1D patients

This finding supports the notion that diabetes is a chronic inflammatory condition with potential long-term adverse effects on kidney health. Urinary cytokines (uMCP-1/Cr, uKIM-1/Cr, uNGAL/Cr) were significantly elevated in T1D patients compared with non-diabetic controls, regardless of disease duration (Table 2). This finding supports that diabetes is a chronic inflammatory condition with potential long-term adverse effects on kidney health. Cytokine levels did not correlate with BMI, which remained within normal age-specific ranges across all groups. Interestingly, patients with long-duration T1D appeared to have lower uMCP-1 concentrations than those with shorter duration (p<0.05).

Urinary cytokine levels were significantly higher in T1D patients with albuminuria (UACR ≥ 30 mg/g) compared with those without albuminuria (Table 3). However, in binary logistic regression analyses adjusted for HbA1c, age, sex, BMI, eGFR, and diabetes duration, urinary cytokines (uMCP-1/Cr, uKIM-1/Cr, and uNGAL/Cr) were not independently associated with albuminuria (all p > 0.05). Higher uNGAL/Cr levels and female sex showed weak trends toward lower odds of albuminuria, but these did not reach statistical significance after adjustment. In contrast, HbA1c consistently demonstrated a strong association with albuminuria, with odds ratios ranging from 0.63 to 0.67 (p <0.001). The overall models were statistically significant (likelihood ratio χ² p <0.0001) and correctly classified approximately 90% of cases (Supplementary Table S2).

  These findings indicate that glycemic control, as reflected by HbA1c, remains the principal determinant of albuminuria in this cohort, whereas urinary cytokines did not independently predict albuminuria status after adjustment for covariates. Furthermore, no significant sex differences in urinary cytokine levels were observed (Supplementary Table S3).

  • Table S2. Pearson correlation coefficients among renal biomarkers, markers of glucose metabolism, and metabolic and hematological parameters in the T1D-S, T1D-M, and T1D-L groups.  (Supplementary Table S2)
  • Table S3. Logistic regression analyses of urinary cytokines (uMCP-1/Cr, uKIM-1/Cr and uNGAL/Cr) and covariates associated with albuminuria. (Supplementary Table S3, newly added)
  • Table S4. Pearson correlation coefficients among renal biomarkers, markers of glucose metabolism, and metabolic and hematological parameters in male and female subjects. (Supplementary Table S4)

  1. Metabolite Interpretation and Biological Relevance:

The discussion (Section 4, lines 357–398) provides a good overview of the roles of NAC, NAA, NAO, and pyruvic acid but lacks depth in linking these metabolites to specific pathophysiological mechanisms in T1D and DKD. For example, the role of pyruvic acid in carbohydrate metabolism is mentioned, but its specific contribution to renal inflammation or tubular injury is unclear. Expand the discussion to include mechanistic insights, supported by additional references if necessary.

The manuscript claims that NAC, NAA, and NAO are “closely linked to T1D progression” (line 42), but the cross-sectional design limits causal inferences. Revise such statements to reflect the study’s observational nature and suggest longitudinal studies to confirm causality.

Reply:

Thank you for these valuable suggestions. Accordingly, we make the revisions as shown in the following:  

  • Revision: Discussion (Line 410-426) and added new references 30, 31,

In our study, pathway analysis revealed crucial alterations in amino acid and     carbohydrate metabolism associated with T1D. Notably, pyruvic acid emerged as a    central metabolite within branched-chain amino acid pathways, linking it to β-cell function, insulin resistance, diabetes, metabolic syndrome, and obesity [26-28]. Beyond its role in carbohydrate metabolism, emerging evidence suggests that pyruvate con-tributes to the pathophysiology of DKD. Under hyperglycemia, impaired glycolysis, mitochondrial dysfunction, oxidative stress, and activation of collateral glucose path-ways (e.g., the polyol pathway) promote tubular injury and inflammation. Pyruvate may counteract these processes by sustaining glycolysis–TCA cycle flux, preserving ATP production, and scavenging reactive oxygen species, thereby limiting oxidative stress and suppressing poly(ADP-ribose) polymerase (PARP) activation [29]. In db/db (diabetic) mice, exogenous pyruvate supplementation restored the activity of          metabolic enzymes (pyruvate kinase and pyruvate dehydrogenase), improved the NAD+/NADH balance, and reduced oxidative stress, advanced glycation end-product (AGE) accumulation, and mesangial expansion, thereby delaying nephropathy           progression [30]. These findings suggest that pyruvate may modulate DKD development through mitochondrial protection, redox regulation, and inhibition of maladaptive glucose pathways.

  • Revision: Abstract (Line 41-46)

Conclusions: Urinary biomarkers (MCP-1/Cr, NGAL/Cr, and KIM-1/Cr) are sensitive indicators of subclinical kidney dysfunction in T1D patients, often preceding albuminuria. Alterations in amino acid-related metabolites (NAC, NAA, and NAO) and pyruvate high-light possible metabolic disturbances associated with T1D duration and oxidative stress. However, given the cross-sectional design, longitudinal studies are needed to confirm causality and clarify their predictive value in DKD progression.

  1. Figure and Table Presentation:

Table 1 (line 201): The table is comprehensive but difficult to read due to dense formatting. Consider splitting it into multiple tables (e.g., demographic, glycemic, lipid, and renal parameters) to improve readability. Additionally, clarify the meaning of “NS” (not significant) and provide exact p-values where possible.

Reply:
Thank you very much for this helpful suggestion.

To improve the readability of Table 1, we have revised its presentation. Specifically, the urinary cytokine data have been removed from Table 1, as these are already reported in Table 2.  Accordingly, the revised manuscript now presents two separate tables: Table 1 and Table 2. In addition, exact p-values have been provided for each comparison in Table 1.

  1. Methodological Details for Metabolomics:

The metabolomic analysis (Section 2.3, lines 109–120) describes the use of 1H-NMR spectroscopy and MetaboAnalyst but lacks specifics on quality control measures (e.g., batch effects, normalization procedures) and the rationale for selecting specific metabolites (e.g., VIP score ≥ 1.0). Provide a more detailed description of the metabolomic pipeline, including quality control steps and criteria for metabolite selection.

The use of both intelligent and variable size bucketing (line 117) is mentioned but not justified. Explain why both techniques were used and how they impacted the results.

Reply:

For downstream analysis in MetaboAnalyst, we used variable importance in projection (VIP) scores from PLS-DA, with metabolites having VIP ≥ 1.0 considered significant contributors. This widely applied threshold reflects features that contribute more than the average to model discrimination. To ensure robustness, differential metabolites were further confirmed by univariate testing (Mann–Whitney, raw p < 0.05) and validated using Random Forest modeling with Boruta feature selection and cross-validation.

In addition, we employed a dual approach combining intelligent bucketing and variable-size binning for spectral preprocessing. Standard equidistant binning may obscure small peaks or split peaks across bins, leading to information loss. Intelligent bucketing allows bin boundaries to adapt to local spectral features, improving the capture of peaks in crowded regions. Variable-size (adaptive intelligent) binning further refines this by recursively adjusting bin edges according to peak shape, thereby minimizing artifacts from small chemical shift variations and avoiding arbitrary bin definitions. Using both methods together ensured accurate alignment of metabolite peaks across spectra and improved classification performance in multivariate models.

Revision section 2.3. (Line 119-144)

2.3.1H- nuclear magnetic resonance (NMR) spectroscopy analysis

Urine samples were prepared and analyzed according to a previously described protocol [22]. Briefly, urine samples (900 μL) were mixed with phosphate buffer in deuterium water (100 μL), containing 0.04% TSP (3-(trimethylsilyl)-propionic-2,2,3,3-d4 acid sodium salt) for chemical shift standardi-zation. After centrifugation at 12,000g for 30 minutes at 4°C, 600 μL of the superna-tant was analyzed using a Bruker Avance 600 MHz NMR spectrometer (Bruker-Biospin GmbH, Karlsruhe, Germany). Spectral preprocessing was   perfom-red with TopSpin v3.2 (Bruker BioSpin, Rheinstetten, Germany) and NMRProcFlow v1.4 [23], including baseline correction, chemical shift calibration, solvent peak exclu-sion, and spectral alignment to minimize variability due to pH, ionic strength, or tem-perature differences. Spectral quality was assessed by signal-to-noise ratio, and noisy or inconsistent regions were excluded. To account for concentration differences and reduce potential batch effects, bucket intensities were normalized before multivariate analysis.

For spectral data reduction, both intelligent bucketing and variable-size (adaptive) binning were applied [24]. Intelligent bucketing allows bin edges to adapt to local spectral features, reducing the risk of splitting narrow peaks or merging adjacent peaks. Variable-size binning further refines this process by recursively adjusting bin        boundaries according to peak shape, thereby minimizing artifacts from small chemical shift variations and improving alignment across spectra. The combined use of these          methods ensured optimal capture of biologically relevant spectral information.

Metabolite identification was performed using Chenomx NMR Suite v8.6. Data were logarithmically transformed, mean-centered, and Pareto-scaled prior to analysis in MetaboAnalyst v5.0. Multivariate analyses included partial least squares          discriminant analysis (PLS-DA), with metabolites having variable importance in projection (VIP) scores ≥ 1.0 considered significant contributors to group separation.

Minor Issues

  1. Terminology Consistency:

The manuscript uses “uMCP-1,” “uKIM-1,” and “uNGAL” interchangeably with “MCP-1/Cr,” “KIM-1/Cr,” and “NGAL/Cr.” Standardize terminology (e.g., consistently use “uMCP-1/Cr” for creatinine-normalized values) to avoid confusion.

Reply:

Thank you for this suggestion.

We have standardized the terminology in the revised manuscript and now consistently use “MCP-1/Cr,” “KIM-1/Cr,” and “NGAL/Cr” to replace “uMCP-1,” “uKIM-1,” and “uNGAL.”

  1. Define abbreviations (e.g., PLS-DA, VIP) the first time they are used in the text, not just in the abbreviations section.

Reply:

We appreciate this reminder. All abbreviations, including PLS-DA and VIP, are now defined at their first appearance in the manuscript.

3.Typographical and Formatting Errors:

Line 88: The term “childhood-onset T1D” should be consistently used instead of “youth-onset” (e.g., line 138) to align with WHO criteria mentioned.

Line 428: “Informed Consent Statement)” appears incomplete and should be corrected to “Institutional Review Board” or similar.

 Reply:

  1. We have corrected the terminology to consistently use “childhood-onset T1D” throughout the manuscript.
  2. The Ethics Statement (Line 501-505) has been revised to clearly include the Institutional Review Board approval (IRB 201601179A3).

4.Reference Formatting:

Some references (e.g., lines 440–542) are incomplete or inconsistently formatted (e.g., missing journal names or DOIs). Ensure all references follow the journal’s citation style and are complete.

 Reply:

Thank you for pointing this out. We have carefully reviewed and corrected all references to ensure completeness and consistency with the journal’s citation style.

5.Clarity in Results:

In Table 4 (line 236), the chemical shift ranges for metabolites (e.g., “2.081–2.092 (s)”) are provided, but the abbreviation “s” (singlet) is not explained in the table legend. Include a clear legend defining all abbreviations (e.g., s, m, t).

Reply:

We have revised the legend of Table 4 (Line 281-284) as follows:

“ Data are reported as chemical shift ranges (δ, ppm). *VIP scores were obtained from partial least-squares discriminant analysis (PLS-DA). †Fold change was calculated by dividing metabolite levels between T1D patients with short-, medium-, and long-term duration. ‡ Only metabolites that met both the conservative significance threshold (p < 0.01) and false discovery rate (FDR) correction (q < 0.05) were considered statistically significant and are shown in bold. Abbreviations: VIP, variable importance in projection; ppm, parts per million; m, multiplet; s, singlet; t, triplet.”

The results section (lines 136–188) is dense and could benefit from subheadings to separate demographic, cytokine, and metabolomic findings for better readability.

Reply:

We agree with this helpful suggestion. The Results section has been reorganized with the following subheadings for clarity:

3.1. Demographic and clinical profiles of T1D groups

3.2. Urinary MCP-1/Cr, KIM-1/Cr, and NGAL/Cr levels in T1D patients

3.3. Distinct urinary metabolic phenotypes and functional pathways of T1D-S, T1D-M, and T1D-L patients

3.4. Relationships between urinary cytokines (uKIM-1/Cr, uMCP-1/Cr, uNGAL/Cr) and  metabolites

Reviewer 2 Report

Comments and Suggestions for Authors

The authors present an interesting and well-structured manuscript. The results are clear, and the scientific approach is sound, but I have minor suggestions:

  • The author should include the ethics approval number in the methods section.
  • The discussion should be expanded to provide a detailed explanation of the mechanistic roles of the studied biomarkers (uKIM-1, uMCP-1, uNGAL) in the pathogenesis and progression of diabetic kidney disease. Linking these mechanisms to the study's findings will improve the work.
  • The authors may also include the limitation of the present findings for a better understanding of the manuscript. 

Author Response

Comments and Suggestions for Authors

The authors present an interesting and well-structured manuscript. The results are clear, and the scientific approach is sound, but I have minor suggestions:

  • The author should include the ethics approval number in the methods section.

Reply:

Thank you very much for this kind reminder. We have included the relevant ethics approval information in the Ethics Statement (Lines 502–506).

  • The discussion should be expanded to provide a detailed explanation of the mechanistic roles of the studied biomarkers (uKIM-1, uMCP-1, uNGAL) in the pathogenesis and progression of diabetic kidney disease. Linking these mechanisms to the study's findings will improve the work.

Reply:

Thank you for this valuable suggestion. We have expanded the Discussion to include detailed explanations of the potential mechanistic roles of KIM-1, MCP-1, and NGAL in DKD pathogenesis and progression, and have added several new references to support these revisions (Lines 447–462).

  • The authors may also include the limitation of the present findings for a better understanding of the manuscript. 

Reply:

We agree with this important point. Accordingly, we have added a detailed description of the study’s limitations in the Conclusion section (Lines 474–484).

Reviewer 3 Report

Comments and Suggestions for Authors

This manuscript presents the metabolomics and cytokine analysis of a cohort of type 1 diabetes patients.  While the datasets and analyses look promising, there are 4 very major issues and 6 major issues in this manuscript that must be addressed.

Very Major:

A) There is no indication that the metabolomics dataset has been deposited into a public scientific repository like MetaboLights or Metabolomics Workbench.  There is not even supplemental material with the dataset, R scripts, and full results.  This is total unacceptable in 21st century science, since it prevents scientific reproducibility and future meta-analyses.  The authors must make the dataset available in a reasonable manner along with the full results of their statistical analyses.

B) No multiple testing correction was performed except for the pathway enrichment analysis.  Table 4 implies that only 14 metabolites were detected and assigned, but were tested across 42 statistical tests. This is a borderline situation with respect to multiple testing correction.  The authors have two options: 1) Apply multiple testing correction and present results in terms of adjusted p-values; or 2) Lower the alpha to 0.01 to prevent false discovery.

C) Should remove the sentence about metabolic pathway analysis from the abstract, since none of the pathways had an FDR below 0.05 or even 0.1.

D) There is no discussion of the potential confounds with respect to age and BMI in the comparisons made to non-diabetic children. This is a serious flaw with the interpretations of uMCP-1, uKIM-1, and uNGAL comparisons.  The controls are clearly not age and BMI matched.

Major Issues:

1) The authors should be more specific about the differences in 4 metabolites highlighted in the abstract.  Which ones had positive or negative correlation (increases or decreases) with disease duration? Also, be more specific with what “significant differences in metabolites” mean? Do the authors mean “statistically significant differences in metabolites”?  If so, what was considered statistically significant in terms of adjusted p-value?

2) The methods and results lack how many metabolites and urinary cytokines were detected and assigned?  Also, what was the level of missingness (non-detection) across the metabolomics and cytokine datasets?  How were missing values handled in the statistical tests and correlations? 

3) There is no justification given for using non-parametric tests.  Use of non-parametric tests is statistically safer, since they do not assume normality.  Simply provide some results in supplemental material that demonstrates the non-normality of the data.  Was the handling of missing values causing non-normality?

4) Something does not read correctly to this reviewer in the paragraph starting at line 158.  How can you have negative coefficients for the cytokine levels to predict albuminuria if these cytokine levels are significantly higher in T1D patients with albuminuria?  Please explain.

5) Figure 3 description in the main text and the Figure 3 title should clearly indicate what correlation (Spearman or Pearson) is represented in the heatmap.

Minor Issues:

Line 39: terms uMCP-1, uNGAL, and uKIM-1 are not defined.  From context, the “u” likely means urinary, but do not make the reader assume.

Author Response

Comments and Suggestions for Authors

This manuscript presents the metabolomics and cytokine analysis of a cohort of type 1 diabetes patients.  While the datasets and analyses look promising, there are 4 very major issues and 6 major issues in this manuscript that must be addressed.5

Very Major:

  1. There is no indication that the metabolomics dataset has been deposited into a public scientific repository like MetaboLights or Metabolomics Workbench.  There is not even supplemental material with the dataset, R scripts, and full results.  This is total unacceptable in 21st century science, since it prevents scientific reproducibility and future meta-analyses.  The authors must make the dataset available in a reasonable manner along with the full results of their statistical analyses.

Reply:

We sincerely appreciate the reviewer’s emphasis on data transparency and reproducibility. According to the regulations of our institution and the stipulations in the informed consent forms signed by participants, there are strict restrictions on sharing personal and experimental data after study completion. Therefore, at this stage, we are bit concerned about placing the full dataset in a public repository.

That said, we fully acknowledge the importance of data accessibility for scientific progress. If required by the journal and reviewers, we will carefully re-evaluate the informed consent documents to explore whether partial deposition (e.g., anonymized or de-identified data) may be feasible within the scope of IRB approval. At that time, we will upload the material to Metabolomics Workbench. Furthermore, as an alternative, the dataset and full statistical results can be made available upon reasonable request to the corresponding author, ensuring that access remains consistent with ethical and legal requirements.

We believe this arrangement appropriately balances the protection of participants’ rights with the need for reproducibility in scientific research.

  1. B) No multiple testing correction was performed except for the pathway enrichment analysis.  Table 4 implies that only 14 metabolites were detected and assigned, but were tested across 42 statistical tests. This is a borderline situation with respect to multiple testing correction.  The authors have two options: 1) Apply multiple testing correction and present results in terms of adjusted p-values; or 2) Lower the alpha to 0.01 to prevent false discovery.

Reply:

We thank the reviewer for this valuable comment.

In response, we have revised the Methods (Section 2.5. Line 225-227) to clarify our statistical approach. Specifically, we now state that metabolite comparisons were corrected for multiple testing using the Benjamini–Hochberg FDR method (q < 0.05 considered significant), and we also applied a conservative alpha threshold of p < 0.01 to further minimize the risk of false discovery. Importantly, four metabolites (N-acetylcysteine [NAC], N-acetylaspartate [NAA], N-delta-acetylornithine [NAO], and pyruvate) remained significant under both criteria, consistent with our original interpretation. This dual approach ensures that our findings are robust and not dependent on a single statistical threshold.

Accordingly, the revision is also made in Abstract (Line 35-41):

Results: Urinary MCP-1/Cr, KIM-1/Cr, and NGAL/Cr levels were significantly elevat-ed in T1D patients compared with non-diabetic controls, but did not correlate with disease duration. Metabolomic profiling identified distinct urinary signatures across T1D duration. Specifically, N-acetylcysteine (NAC) and N-delta-acetylornithine (NAO) increased progressively, while N-acetylaspartate (NAA) and pyruvic acid decreased with longer disease duration. These four metabolites remained statistically significant after both based on Mann–Whitney tests with false discovery rate (FDR) correction (q < 0.05) and application of a conservative alpha threshold (p < 0.01), suggesting potential disruptions in amino acid and carbohydrate metabolism.

  1. C) Should remove the sentence about metabolic pathway analysis from the abstract, since none of the pathways had an FDR below 0.05 or even 0.1.

Reply:

We thank the reviewer for this helpful comment.

We agree that pathway analysis did not yield statistically significant results after FDR correction (FDR < 0.05 or 0.1). Therefore, we have removed the sentence about pathway analysis from the abstract: “Additionally, metabolic pathway analysis highlighted notable alterations in amino acid and carbohydrate metabolism.

Instead, we now restrict pathway interpretation to the Discussion, where we describe the potential biochemical pathways underlying the significant metabolites identified (e.g., NAC, NAA, NAO, pyruvate). This ensures that pathway-level findings are not overstated, while still providing biological context for the observed metabolite changes.

  1. D) There is no discussion of the potential confounds with respect to age and BMI in the comparisons made to non-diabetic children. This is a serious flaw with the interpretations of uMCP-1, uKIM-1, and uNGAL comparisons.  The controls are clearly not age and BMI matched.

Reply:

We thank the reviewer for this important comment. We acknowledge that the non-diabetic controls were not perfectly matched for age and BMI across all T1D subgroups, largely due to limitations in research funding and the feasibility of recruiting a large control cohort. This is a limitation of our study and is now noted in the revised Discussion.

Nevertheless, several points support the robustness of our findings:

  • Cytokine elevation observed even in the youngest group: Urinary MCP-1, KIM-1, and NGAL were already significantly elevated in the short-duration T1D group (T1D-S) compared with age-matched non-diabetic controls. This suggests that diabetes itself, rather than age or BMI differences, is the primary driver of increased tubular injury marker expression.
  • BMI distribution: BMI values remained within age-appropriate reference ranges in all T1D subgroups and controls, minimizing the likelihood that BMI alone explains the observed differences.
  • Expected trends with disease duration: It is reasonable to assume that if these markers are elevated in T1D-S compared to age-matched controls, higher levels would also be expected in the medium- and long-duration groups, consistent with the chronicity of tubular stress in diabetes.

We have clarified this issue in the Discussion and highlighted that future studies with larger, age- and BMI-matched control groups are needed to fully exclude residual confounding.

Revision:

Conclusion: (Line 473-483)

Our findings elucidate intricate interactions between metabolic disturbances and inflammation in DKD associated with T1D. Urinary biomarkers (uKIM-1/Cr, uMCP-1/Cr, uNGAL/Cr) and specific metabolites show promise for early identification of subclinical kidney dysfunction, even preceding albuminuria. These characteristic metabolic signatures, particularly those linked to branched-chain amino acid metabo-lism, may provide valuable insights into disease progression; however, given the cross-sectional design and the lack of perfectly age- and BMI-matched controls, causal relationships cannot be inferred. Longitudinal studies with larger, matched cohorts are warranted to validate these observations and to clarify whether these metabolites          directly contribute to, or result from, T1D progression. Stratification of participants by clinical characteristics or disease stage will further strengthen future research.

Major Issues:

  • The authors should be more specific about the differences in 4 metabolites highlighted in the abstract.  Which ones had positive or negative correlation (increases or decreases) with disease duration? Also, be more specific with what “significant differences in metabolites” mean? Do the authors mean “statistically significant differences in metabolites”?  If so, what was considered statistically significant in terms of adjusted p-value?

Reply:

We thank the reviewer for this important comment.  In our study, N-acetylcysteine (NAC) and N-delta-acetylornithine (NAO) showed progressive increases with longer T1D duration, while N-acetylaspartate (NAA) and pyruvic acid exhibited decreases from short- to long-duration T1D. By “significant differences in metabolites,” we refer to statistically significant group differences as assessed by Mann–Whitney tests, with multiple testing correction applied using the false discovery rate (FDR). Metabolites were considered statistically significant when they met the criteria of raw p < 0.05 and FDR-adjusted q < 0.05. This clarification has now been added to Abstract (Line 38-40), Methods (2.5. Statistical analyses, Line 163-167), and Results (Line 225-227) to ensure consistency and transparency.

  • The methods and results lack how many metabolites and urinary cytokines were detected and assigned?  Also, what was the level of missingness (non-detection) across the metabolomics and cytokine datasets?  How were missing values handled in the statistical tests and correlations?

Reply:

We thank the reviewer for raising this point. In our study, complete datasets were available for all participants. Specifically, 1H-NMR analysis generated metabolomic profiles for 247 urine samples, and urinary cytokines (uMCP-1, uKIM-1, uNGAL) were successfully measured in the same 247 participants. There were no missing or non-detected values across either dataset; therefore, no imputation or special handling of missing values was required. All statistical tests and correlation analyses were performed on complete cases. The number of metabolites identified and annotated is presented in Section 2.3 and Table 4, and the cytokine markers are summarized in Section 2.2 and Table 2.

3) There is no justification given for using non-parametric tests.  Use of non-parametric tests is statistically safer, since they do not assume normality.  Simply provide some results in supplemental material that demonstrates the non-normality of the data.  Was the handling of missing values causing non-normality?

Reply:

  • We thank the reviewer for raising this important point.

In our study, Kruskal–Wallis ANOVA followed by Dunn’s post hoc tests (non-parametric tests) were selected because many of our variables did not follow a normal distribution, as confirmed by Shapiro–Wilk tests (OriginPro). This non-normality persisted even after attempted log-transformation. We will include representative distribution plots and normality test results as Supplementary Table S1 to demonstrate this.

  • There are no missing data in this analysis.

4) Something does not read correctly to this reviewer in the paragraph starting at line.  How can you have negative coefficients for the cytokine levels to predict albuminuria if these cytokine levels are significantly higher in T1D patients with albuminuria?  Please explain.

Reply:

We appreciate the reviewer’s insightful comment.

The apparent discrepancy arises because two different types of analyses were performed. In original unadjusted comparisons (Table 3), urinary MCP-1/Cr, KIM-1/Cr, and NGAL/Cr levels were indeed significantly higher in T1D patients with albuminuria compared to those without, reflecting their role as markers of tubular injury and inflammation. However, in the multivariable binary logistic regression, we adjusted for key covariates (HbA1c, age, sex, BMI, eGFR, and diabetes duration). After adjustment, the cytokine associations with albuminuria were attenuated and the coefficients turned negative, indicating that their effect was largely explained by other factors, especially glycemic control. In fact, HbA1c consistently remained a strong and independent predictor of albuminuria (OR ~0.63–0.67, p < 0.001), while urinary cytokines lost independent significance.

This does not imply that higher cytokine levels are protective, but rather that, once shared variance with HbA1c and other covariates is accounted for, the independent predictive contribution of cytokines diminishes and the regression coefficient may become negative due to collinearity or confounding. This highlights the central role of poor glycemic control in driving albuminuria in our cohort, while elevated cytokines still reflect underlying tubular stress and inflammation, but not as independent predictors when modeled alongside HbA1c and other clinical factors

Revision:

  1. Table 3 (Line 264)
  2. Result (Line 195-210)

Urinary cytokine levels were significantly higher in T1D patients with albuminu-ria (UACR ≥ 30 mg/g) compared with those without albuminuria (Table 3). However, in   binary logistic regression analyses adjusted for HbA1c, age, sex, BMI, eGFR, and diabetes duration, urinary cytokines (uMCP-1/Cr, uKIM-1/Cr, and uNGAL/Cr) were not independently associated with albuminuria (all p> 0.05). Although urinary      cytokine levels were higher in albuminuric patients, their independent predictive value diminished in multivariable models after adjustment for HbA1c and other covariates, reflecting confounding and collinearity rather than a protective effect. In contrast, HbA1c consistently demonstrated a strong association with albuminuria, with odds ratios ranging from 0.63 to 0.67 (p<0.001). The overall models were statistically significant (likelihood ratio χ² p<0.0001) and correctly classified approximately 90% of cases (Supplementary Table S3).

These findings indicate that glycemic control, as reflected by HbA1c, remains the prin-cipal determinant of albuminuria in this cohort, whereas urinary cytokines did not in-dependently predict albuminuria status after adjustment for covariates. Furthermore, no significant sex differences in urinary cytokine levels were observed (Supplementary Table S4).

5) Figure 3 description in the main text and the Figure 3 title should clearly indicate what correlation (Spearman or Pearson) is represented in the heatmap.

Reply:

Thank you for this suggestion. We have revised the legend of Figure 3 (Lines 394–397) to clarify the correlation method used:

Figure 3. Spearman correlation heatmap of characteristic urinary metabolites, tubular inflammatory cytokines (urinary NGAL/creatinine, MCP-1/creatinine, and KIM-1/creatinine), renal function markers (eGFR, urine ACR, BUN, serum creatinine), and blood biochemicals (HbA1c, glucose, uric acid) in 247 patients with childhood-onset T1D.

Minor Issues:

Line 39: terms uMCP-1, uNGAL, and uKIM-1 are not defined.  From context, the “u” likely means urinary, but do not make the reader assume.

Reply:

Thank you very much for this reminder. To improve clarity and avoid reader assumptions, we have revised the terminology so that uMCP-1, uNGAL, and uKIM-1 are now consistently written as MCP-1, NGAL, and KIM-1 throughout the manuscript.